# Serologic Tools and Strategies to Support Intervention Trials to Combat Zika Virus Infection and Disease

**DOI:** 10.3390/tropicalmed4020068

**Published:** 2019-04-19

**Authors:** Matthew H. Collins

**Affiliations:** Hope Clinic of the Emory Vaccine Center, Division of Infectious Diseases, Department of Medicine, School of Medicine, Emory University, Decatur, GA 30030, USA; matthew.collins@emory.edu; Tel.: +1-404-712-8942

**Keywords:** Zika, virus, vaccine, arbovirus, vector-borne disease, serology, diagnostics, cross-reactivity, clinical trial

## Abstract

Zika virus is an emerging mosquito-borne flavivirus that recently caused a large epidemic in Latin America characterized by novel disease phenotypes, including Guillain-Barré syndrome, sexual transmission, and congenital anomalies, such as microcephaly. This epidemic, which was declared an international public health emergency by the World Health Organization, has highlighted shortcomings in our current understanding of, and preparation for, emerging infectious diseases in general, as well as challenges that are specific to Zika virus infection. Vaccine development for Zika virus has been a high priority of the public health response, and several candidates have shown promise in pre-clinical and early phase clinical trials. The optimal selection and implementation of imperfect serologic assays are among the crucial issues that must be addressed in order to advance Zika vaccine development. Here, I review key considerations for how best to incorporate into Zika vaccine trials the existing serologic tools, as well as those on the horizon. Beyond that, this discussion is relevant to other intervention strategies to combat Zika and likely other emerging infectious diseases.

## 1. Introduction

The recent Zika virus (ZIKV) epidemic was declared an international public health emergency in early 2016 and has caused over 3500 birth defects since its emergence in the Americas [1]. The full spectrum of disease and sequelae due to congenital ZIKV infection is still being defined, but it appears that approximately 5–11% or more of ZIKV-infected pregnant women have had their pregnancies complicated by one of several adverse fetal outcomes (AFO) collectively termed congenital Zika syndrome (CZS) [2,3]. Additionally, ZIKV has exhibited unique phenotypes compared to other flaviviruses including sexual transmission [4] and neurologic sequelae [5,6,7]. Despite the potential to cause catastrophic disease, the majority of ZIKV infections are inapparent (very mild or completely asymptomatic), as is true for many arboviral infections [8,9,10,11]. However, inapparent infections are not unimportant infections. For the related dengue virus (DENV), asymptomatic infections are major contributors to ongoing transmission [12,13]. With ZIKV, asymptomatic infections can lead to sexual transmission [14] or cause birth defects when occurring in a pregnant woman [15,16].

ZIKV belongs to the genus *flavivirus*, which includes several medically important viruses, such as the yellow fever virus, West Nile virus, Japanese encephalitis virus, and DENV. This group of viruses has positive-sense, single-stranded RNA genomes that code for a single polypeptide that is further processed in three structural and five non-structural proteins [4]. Like DENV, yellow fever, and chikungunya (an alphavirus), ZIKV is transmitted primarily though the bite of an infected *Aedes (Stegomyia) aegypti* mosquito [17]. This vector thrives in urban and peri-urban environments, making it the ideal vector for propagating epidemic human transmission [18,19,20,21]. Furthermore, due to the vast endemic range that includes most of the tropics and many areas of high population density, billions of people are at risk for at least one flavivirus infection, with (DENV) alone causing approximately 400 million infections annually [9,22]. In the recent ZIKV epidemic, attack rates were high in certain regions, up to 73% in northeast Brazil [23], with seroprevalence now exceeding 50% in multiples areas [24,25]. However, there are likely over a billion Zika-susceptible individuals living in areas where competent Zika vectors are endemic [20,21,26,27], making future outbreaks highly likely.

The emergence of ZIKV in Latin America, which occurred primarily in DENV-endemic regions, highlighted critical requirements for a range of tools and strategies to control ZIKV infection. Development of a safe and effective ZIKV vaccine has been a top priority since the beginning of the outbreak in Brazil [19,28,29]. Central to vaccine development and most public health efforts to combat ZIKV is the need for serologic assays with the capacity to discriminate between ZIKV and DENV (or other flavivirus) infections [30,31]. Considerable scientific advances have provided important background knowledge on the humoral immune response to ZIKV. These advances include analyses of serologic responses [32,33,34,35,36,37,38] and B cells and monoclonal antibodies (Abs, mAbs) [39,40,41,42,43,44,45,46,47] as well as evaluation of diagnostic kits and assays [48,49,50,51,52,53]. However, no commercial serodiagnostic test is available outside of emergency use authorization by the FDA [54], and no recently developed serologic assay has emerged as gold standard to be broadly implemented in healthcare and research settings. Thus, a critical challenge now facing the field is to hone and harmonize serologic approaches for diagnosing recent and remote ZIKV infection, regardless of clinical manifestations, to support clinical care, surveillance, and clinical trials to assess interventions, such as vaccination, to reduce incident ZIKV infection.

## 2. Objective

In this article, I review the challenges surrounding the serologic diagnosis of ZIKV infection, discuss progress made in the field in the last few years, and suggest a core set of considerations to serve as a resource for those designing intervention trials to reduce ZIKV infection and transmission. The themes explored here should also be relevant for those conducting surveillance and clinical diagnostic testing of ZIKV for public health or patient care purposes. Ultimately, many of these concepts should apply to public health responses to other arboviruses, potentially the next emerging infectious disease, as ZIKV is part of a recurring phenomenon, by which additional pathogenic RNA viruses are likely to emerge [55,56,57].

## 3. Natural History of ZIKV Infection, Host Immune Response, and Serologic Diagnosis

Human infection by ZIKV was described in very few reports until physicians on Yap Island along with the US Center for Disease Control reported an outbreak in 2007 [58]. We now know that ZIKV infection is frequently inapparent, with approximately 20% of those infected developing a self-limiting illness most often characterized by rash, fever, conjunctivitis, and/or arthralgia/myalgia [58,59,60,61]. The incubation period is presumed to be less than one week, and symptom duration is less than one week in most cases [61]. Viremia is typically cleared quickly after symptom onset, but the infectious virus has been isolated from semen for several weeks after infection [62,63,64], and ZIKV RNA can be detected in blood or vaginal secretions for prolonged times, particularly in pregnancy [65,66,67], and for at least 6 months in semen, following infection [62,68,69,70,71,72]. Microcephaly and other birth defects and neurodevelopmental problems are the most concerning manifestations of ZIKV infection when it occurs during pregnancy and is vertically transmitted to the fetus [2,3,73,74,75]. Additionally, an increased incidence of Guillain-Barré syndrome has been consistently noted after Zika outbreaks in multiple countries [5,6,7]. Other complications and severe outcomes have been very rarely reported, ranging from low platelets to death, usually in patients with additional comorbid factors that may have contributed to the pathogenicity of their illness [76,77,78,79].

The diagnosis of ZIKV and many arbovirus infections can be accomplished by molecular or serologic methods [80,81]. The former requires direct evidence for the infecting virus from a clinical specimen, such as viral isolation through cell culture, specific amplification of viral genomic material by reverse transcriptase–polymerase chain reaction (RT-PCR), or detection of antigens produced during viral replication [82]. Molecular diagnostics, particularly RT-PCR, have the advantage of being highly specific for ZIKV and can be performed on noninvasive specimens, such as urine and saliva. However, sensitivity may decline as early as a few days post onset of symptoms (DPO), and RT-PCR is generally not used beyond 14 DPO [83]. This method is not appropriate for the surveillance or detection of inapparent or prior infection, with the exception that RT-PCR testing is recommended for detecting inapparent ZIKV infection during pregnancy when there is ongoing exposure [84].

Serologic diagnostics are considered indirect, as these assays detect the antibody response elicited by a recent or remote infectious event [80,81,84]. Human flavivirus infection elicits a complex antibody (Ab) response, which also has a central role in immunity and pathogenesis [85,86,87,88]. A detailed understanding of the human Ab response to ZIKV still awaits ongoing and future research; however, key features have been defined or extrapolated from extensive experience with closely related viruses. ZIKV-reactive IgM is detectable within 4–7 days of symptom onset. ZIKV IgM testing is useful for diagnosing symptomatic infections and recent asymptomatic infections [83]. IgM testing remains useful for diagnosing CZS but is no longer recommended in evaluating asymptomatic pregnant women in non-endemic areas with potential exposure to ZIKV [84]. A positive test is supportive but not definitive for ZIKV infection, and confirmatory neutralization may be required [83,84]. The duration of the anti-ZIKV IgM response is unclear, but it can persist beyond 12 weeks in some cases [83,89,90], which limits the reliability of this assay to narrowly define a recent ZIKV infection. 

IgG against ZIKV becomes detectable by 10–14 days and presumably lasts for years, as with other flaviviruses [81,83,85,90]. The kinetics of the IgG response have direct bearing on performance of serologic testing. Assays that detect IgG may not reach peak sensitivity until after the first 10 to 15 DPO [91]. Additionally, the magnitude of cross-reactive Ab and their relative abundance may be highest early after infection [37,85,92], compromising assay specificity until the late convalescent period. IgG testing is not part of CDC recommendations for ZIKV diagnosis other than its potential contribution to neutralization activity [83,84]. However, IgG detection forms the basis for many diagnostic assays under development and the IgG response is central to understanding the immunity elicited by vaccination and natural infection. The targets of these Abs include structural proteins as well as nonstructural proteins (ns), most notably NS1 [40,93]. One hundred eighty envelope protein (E) monomers decorate the surface of the ZIKV virion, organizing into head-to-tail homodimers, which further arrange into higher order structures that give a herringbone appearance and icosahedral symmetry [94,95]. ZIKV and DENV E are approximately 50% conserved [32,40,94], but the homology is not homogenous; the fusion loop of E domain II is highly conserved, whereas E domain III is the most divergent and may be more likely to be targeted by ZIKV-specific Ab responses [40,46,92,93,96]. In addition to epitope, flavivirus-elicited Abs also vary in terms of their specificity, kinetics, and function. Abs may be very specific for one virus or cross-react to two or more viruses [87,97,98]. Cross-reactivity that limits the specificity of serologic assays is the major challenge for this method of diagnosis [30,31,87,89]. It is also worth noting that tests that detect viral antigens may also be subject to similar problems that complicate serologic tests because a mAb is typically used to capture the antigen of interest from a biospecimen.

A subset of binding Abs also exhibit neutralization properties, and it has been shown that neutralizing Abs (nAbs) tend to bind serotype-specific epitopes that often require the three-dimensional structural integrity of the virus particle [85,99,100,101,102,103,104]. In DENV infection, poorly neutralizing, cross-reactive Abs, frequently binding to epitopes in precursor membrane protein (PrM) or the fusion loop [85,93], are implicated in the pathophysiology of severe diseases via Ab-dependent enhancement (ADE) [105,106]. There remains concern that non-neutralizing cross-reactive Ab elicited by prior DENV infection may exacerbate ZIKV infection or potentiate vertical transmission [107,108,109,110,111,112]; however, there is nonhuman primate data to the contrary [113,114], and no epidemiologic data in humans support the hypothesis [115,116,117]. The possibility of ZIKV infection enhancing subsequent DENV infection is even less well studied. 

## 4. Available Serologic Tests for ZIKV

The assays available for serologic diagnosis of ZIKV infection range from standard lab techniques that have been used for decades to novel assays that leverage innovative technologies. The goal here is not an exhaustive catalogue of every serologic test under development or in use for ZIKV, but to give an idea of the spectrum of assays available and ongoing work to improve these tools to inform the discussion of implementation of serodiagnostics in intervention trials. IgM capture enzyme-linked immunosorbent assay (ELISA) is a staple in arbovirus diagnosis [118] and is available for diagnostic testing of ZIKV infection [37,119]. The performance of this assay, as well as several commercial kits [120] or in-house assays, has been examined. Sensitivities for in-house [121] or commercial tests [50] can be high, reported up to 100%, and one multi-test algorithm took advantage of this high sensitivity to use IgM as a “rule out” test for suspected Zika cases [122]. The utility of IgM in diagnosing ZIKV in the newborn is also noteworthy since the timings of infection and cross-reactivity with other flaviviruses are much less of an issue [84,123]. However, the utility of IgM testing as a stand-alone assay is limited by the cross-reactivity of IgM elicited by DENV infection, reports of persistent IgM months beyond the early convalescent period [83,89,124,125], and the need for additional testing [37,83,126]. Interesting ongoing work suggests that improved specificity can be obtained by isolating the IgM-attributable portion of neutralizing activity for several months after flavivirus infection [127]. 

IgG cross-reactivity among flaviviruses in traditional assays is at least as problematic as with IgM assays [31,86,87]. However, tests based on the ZIKV-specific binding of IgG are more numerous. Strategies have included targeting antigens derived from E protein subunits that are less conserved between DENV and ZIKV [92], incorporating IgG avidity [122,128,129,130], detecting isoforms (IgG3) that are transiently produced following infection [23], assessing signatures of binding to an array peptide antigen [131], establishing multiplex antigen panels [132], and measuring serum Ab competition for epitope sites of highly ZIKV-specific monoclonal Ab [91]. Determining the titer of neutralization activity, which can be due to IgG, IgM or a combination of the two, remains an important strategy. Plaque reduction neutralization testing (PRNT) [126,133] is still widely performed, but is increasingly replaced by focus reduction neutralization testing (FRNT) [22,134] which detects target cell infection by intracellular immunostaining for viral proteins. FRNT has the advantage of faster turn around and being more amenable to the 96-well format compared to PRNT. Flow cytometry-based methods for detecting viral infection have also been validated for neutralization testing [135,136]. The major drawback of neutralization testing is that it requires a culture of a live virus (BSL-2 containment) and may rely on sophisticated analytic equipment, limiting its deployability in resource-limited settings. Further advances in neutralization testing that improve throughput and/or safety are being elaborated [137,138,139,140]. While several promising serologic assays are under development, neutralization testing is likely to endure for at least the near future as the most widely accepted test for detecting ZIKV-specific Ab responses [82,133,141], even with longstanding issues with lab-to-lab comparability [142] and the ongoing need to standardize this assay.

## 5. Interventions to Control Zika Virus, Trial Design, and Outcomes of Interest

Selection of the most appropriate serologic testing strategy depends heavily on the question to be answered. Defining the intervention and how that intervention is assessed are necessary primary considerations. Though many promising candidate ZIKV vaccines have been developed, an approved, effective vaccine against ZIKV remains an unmet goal [143]. Vaccines are a mainstay of public health and represent some of the most successful ventures of modern medicine, and there is strong precedent for development of successful flavivirus vaccines [22,144,145,146,147]. There are more than 40 ZIKV vaccine platforms in various stages of testing, several with favorable results in pre-clinical challenge experiments. Features of the vaccine platform directly affect serologic assessment, as one of the goals of vaccination is to generate a durable protective Ab response in the vaccine [28]. Almost all ZIKV vaccines will elicit Ab against E. However, many vaccines do not include nonstructural protein antigens, such as NS1. Thus, no anti-NS1 Abs will be elicited, making serologic methods based on detecting anti-ZIKV NS1 an attractive means to distinguish between Ab responses induced due to vaccination versus natural infection. Platforms in this category include an inactivated whole virus [148], virus like particles (VLPs), E subunit vaccines [149,150], ZIKV surface antigen expression in replicating viral vectors [151,152], or induced expression of ZIKV E or VLPs through delivery of mRNA [153] or DNA [154,155]. Passive immunization is related to vaccination but would confer protection for a limited time interval and not long-term immunity, for example protection against ZIKV infection or vertical transmission by infusion of nAb in the period surrounding pregnancy [44].

The most robust trial for assessing efficacy of a vaccine is a randomized, placebo-controlled, double-blinded clinical trial [156], and borrowing from DENV vaccine trial guidelines, virologically confirmed infection is the preferred primary outcome [157]. If the same is followed for ZIKV, assessing this outcome requires virus isolation, detection of ZIKV-specific genomic RNA via a PCR-based assay [158,159], or detection of ZIKV NS1 protein [160] in a subject experiencing an illness meeting the trial’s case definition. Expert panels have worked to harmonize the assessment of additional clinical endpoints in DENV vaccine trials [161]. Published guidelines also advocate for power calculations to be based on confirmed case numbers [157]. In a highly controlled setting, researchers have used virologic measures (viremia) to define the primary outcome in Phase III trials for a DENV vaccine tested in the human challenge model and assessed clinical endpoints as secondary outcomes [162]. The intensive and strictly timed sampling schema for this kind of endpoint would not be feasible in a large field trial. Interestingly, the DENV guidelines relegate serologic endpoints to the assessment of secondary outcomes or subgroup analyses, emphasizing shortcomings in specificity of serologic assays that lead to uncertainty in case classification [157]. However, because symptomatic illness is necessarily a subset of infections, serologic endpoints that assess total infection may have several advantages including efficiency of trial implementation related to monitoring and sampling intensity of the study population.

In addition to vaccine development, recent clinical trial initiatives to assess vector control interventions to reduce arboviral disease have also been gaining traction [163,164,165,166]. The interventions include direct killing of mosquitos via trapping, use of insecticidal chemicals [18], or deploying mosquito predators, such as larvivorous fish [167], to ecological interventions, such as education and environmental management [168,169]. Novel strategies that include biologic or genetic modification of mosquito vectors to reduce or replace the population of competent vectors in a confined geographic area are also recently being tested in observational field studies and clinical trials [163,164,170,171]. Assessing ZIKV infection rates by serologic assays in vector control studies should be similar to, if not less challenging than, in a traditional vaccine trial, since discriminating between seropositivity due to vaccination vs natural infection would not be required. 

Relevant trial designs that are being used, particularly to study vector control interventions, include the innovative test-negative design. An example of this approach is the Applying Wolbachia to Eliminate Dengue (AWED) study ongoing in Indonesia [163]. This trial is a parallel, two-arm, non-blinded cluster randomized controlled trial. The outcome of interest is febrile illness due to DENV, which is assessed by laboratory testing of symptomatic individuals presenting to local health centers. The symptomatic subjects that test negative for arboviral disease (i.e., have an alternative etiology for their febrile illness) serve as internal controls, since the rate of febrile illness due to pathogens not transmitted by *Aedes aegypti* should be unaffected by the intervention. Thus, the odds ratio for living in a treatment area among DENV cases vs test-negative controls defines efficacy. Additional designs include step-wedge, before and after intervention studies and cluster randomized controlled trials (CRCTs) [172]. CRCTs [168,173] are being implemented with use of typical endpoints of incident rates of either laboratory-confirmed disease or total infection (includes inapparent and apparent infections).

## 6. Selection and Optimal Implementation of Serologic Tests in ZIKV Intervention Trials 

Serologic tests serve two principal purposes in assessing impact of interventions to reduce ZIKV transmission: (1) Measuring the rate of overall infection, particularly inapparent infection [23,30], by detecting changes in serologic status across sequential time points that reflect interval infection, and (2) supporting molecular diagnostics for symptomatic ZIKV cases by extending sensitivity of laboratory testing and refining specificity. Viremia and NS1 antigenemia may be transient, leading to decreasing sensitivity with each day post onset of symptoms. Conversely, ZIKV RNA detection may persist beyond acute infection [174,175] or be present in asymptomatic individuals [176]. In both contexts, detecting increasing levels of IgM and/or IgG in convalescent compared to acute serum samples in a subject with a clinical syndrome consistent with arbovirus infection is strongly suggestive of an acute flavivirus infection. This approach also mitigates against misclassification of cases by using a single ZIKV IgM test, since IgM may persist for several months [83,89,177]. Convalescent sampling of suspected cases can present logistical challenges, but the gains in data quality make this activity critical to include whenever feasible.

The key decision points are based on three questions: (1) What is the mechanism of the intervention? (2) What is the goal of the intervention? (3) What is the primary outcome of interest? If the intervention is a vaccine, protection occurs primarily at the individual level (though herd immunity can protect unvaccinated members of a population or those with poor response to the vaccine). The mechanism of protection is through induction of immunity, so vaccination may elicit Ab that lead to positive serologic test results. As mentioned above, assessing Ab responses to NS1 may be particularly useful in this context via direct NS1-binding ELISA or the recently described NS1 BOB assay [91]. This assay is based on Ab competition (blockade of binding (BOB)) at a ZIKV-specific epitope on NS1 by Ab in the test serum, and has a sensitivity of 95% after 20 DPO and a specificity of 89–96%. The goal of all interventions, but perhaps more pertinent to vector control interventions, will include reduction of total infections, which may be discernible with interval sampling for seroconversion. Assay specificity may make it challenging to detect seroconversion (from test negative to test positive) in DENV-endemic areas. While crude IgG ELISA to whole ZIKV antigen could be a very efficient tool for screening a large number of samples in a low DENV and ZIKV prevalence setting, highly specific IgG binding assays or neutralization assays would likely be needed in a DENV-endemic area. In this setting, assays that give results across a range of magnitudes or titers may be more useful than tests that only yield binary outcomes. For example, we have observed that neutralization testing retains value in discriminating between the DENV and ZIKV infection when titers are compared [34,178]. A second issue is that Ab responses are dynamic and sampling frequency can affect sensitivity. An analysis of DENV-specific serologic findings in sequential samples suggests that substantial sensitivity is lost if the sampling frequency is less frequent than every 3–6 months [179]. From an efficiency standpoint, a longstanding strategy to increase the throughput of classical neutralization testing is to test a single serum dilution rather than a dilution series [180]. This approach has been effectively employed in large epidemiologic studies [181]. Selecting assay cut offs is not standardized for this approach, but the effectiveness of serostatus determinations with single dilution neutralization testing has been examined [142] and can support researchers in making practical decisions for implementing this approach. The goal of vaccine or other medical intervention trials will likely reflect some clinical or disease state. Serology plays a supportive but crucial role when symptomatic infection or birth outcome is the primary outcome of interest. Molecular testing should be the mainstay of testing subjects with a suspected symptomatic ZIKV infection. The results of molecular testing should then be refined by determining whether acute and convalescent serologic testing is consistent with an acute ZIKV infection. The most salient features affecting implementation decisions are summarized in Table 1. 

## 7. Additional Considerations in Serologic Assessment of ZIKV Intervention Trials

Beyond the primary purpose of diagnosing ZIKV infection, there are additional advantages to the collection, storage, and study of serum specimens that should be considered during trial planning. Particularly in vaccine trials, serologic studies may be used to identify correlates of protection or explore a mechanistic understanding of divergent clinical outcomes. Given recent issues with DENV vaccine development [182,183,184,185], a potential increase in adverse events or illness in the vaccinated group will be closely monitored in any ZIKV vaccine trial. If a safety issue were discovered, a comprehensive serologic assessment of those subjects should be pursued. Archived sera may be tested using novel assays not available at the time of collection that have improved performance in diagnosing; also, such repositories constitute an invaluable resource for additional research. For example, the recent ZIKV epidemic demonstrated the utility of leveraging existing DENV cohort studies to quickly study many aspects of the emergence of ZIKV [186]. Specimens that are collected in a systematic fashion, preferably with ancillary clinical and demographic data, enable robust study of diagnostic assays, epidemiology and immunology questions that arise in the wake of viral emergence, and serologic surveillance of other pathogens, including those that may be unknown or even undiscovered at the time of specimen collection. Also, humans mount Ab responses to not only the viruses they encounter, but also the “foreign” proteins found in insect saliva, and these responses can be quantified [187,188,189]. Using the same clinical specimen to simultaneously study the virus and the vector has clear advantages in cost and efficiency. Thus, operations for the serologic assessment of a study population can provide rich data and specimen sets to understand the effect of an intervention far beyond yielding a binary classification of subjects as susceptible or immune. 

The areas most heavily affected and at the highest future risk for ZIKV outbreaks are low and middle-income countries (LMIC) [190,191]. The low rate of laboratory-confirmed cases early in the ZIKV epidemic in South America [15] illustrates why capacity for testing and type, complexity, and safety of serologic testing should be taken into account. For serologic methods based on Ab binding and colorimetric analysis, there are advantage in deployability. The protocols are typically straightforward and there is no requirement for sophisticated analytic equipment or biosafety containment for viral culture. This allows trial activities to simultaneously build diagnostic capacity in areas where it is most needed.

Because CZS is the major public health problem caused by ZIKV infection, it is vital to consider the involvement of pregnant women in ZIKV intervention trials. With influenza infection being an exception, there is very little literature on Ab-mediated immunity to emerging infectious diseases in pregnant compared to non-pregnant women [192], one reason being that pregnant women are often systematically excluded from research studies [193]. A recent non-human primate study of pregnant versus non-pregnant female macaques suggested no differences, but more work will be required to assure that we understand the effect of the pregnant state on serologic responses ZIKV and whether potential differences affect the performance of serodiagnostic tests.

Given that intervention trials require large study populations to achieve adequate power—typically at least a few thousand subjects—the timing, frequency, and method of sample collection is a critical issue for several reasons, including safety and regulatory, data quality, cost, and field logistics and operations. For measuring the rates of symptomatic ZIKV cases, the timing and frequency is necessarily determined by when a subject experiences a symptomatic illness consistent with the case definition rather than any per protocol study timetable. Large prospective cohorts for DENV indicate that children may experience approximately 0 to 5 febrile episodes that require sample collection and laboratory testing [181,194]. The case definition is an important consideration. Assessing all acute febrile and rash illnesses can be overly burdensome from the standpoint of cost, field operations, and laboratory testing. Conversely, a very detailed case definition may lead to much higher rate of suspected ZIKV cases that are confirmed, but sensitivity may falter, which could lead to insufficient events to achieve adequate statistical power. The case definition is a particular challenge with ZIKV, as several reports indicate that acute cases may present with only one or a few nonspecific symptoms, such as rash, and fever is not always as highly prevalent in ZIKV cases as it is with other arboviruses [59,78,195]. These factors are not trivial for establishing trial feasibility. For example, for a hypothetical trial with confirmed symptomatic ZIKV infection as the primary outcome that enrolls 2000 children with an average of just 1 suspected case per child per year, 4000 specimens would need to be collected and tested by acute and convalescent serology. Multiple serologic tests (IgM, IgG, neutralization testing) for ZIKV may be performed on each specimen, as well as molecular testing on the acute specimen. Serologic tests for other common causes of acute illness in the study population may also be required. Thus, the number of laboratory assays performed in one year will be several fold more than 4000. Clearly, tools that streamline testing without compromising data quality will facilitate completion of large intervention trials.

Finally, the word “serology” implies the need for serum (obtained by phlebotomy of a peripheral vein); however, a few notable alternatives exist. Dried blood spots (DBS) are one such tool with a favorable track record in field studies [196,197,198,199]. Immunoglobulin proteins tend to be stable over time provided that cards are kept dry and give reliable results in most serologic assays. DBS samples are easier to obtain, process, store, and ship compared to phlebotomy samples. The main drawback to DBS is sample volume, limiting the extent of testing and storage of specimens for future use. Use of oral fluids (saliva) as a specimen for serologic assays has distinct advantages in safety (no needles are required) and patient acceptability and has been effectively used for several infectious diseases [200,201,202]. These assays are also amenable to point-of-care development, which would further optimize field and laboratory operations.

## 8. Conclusions

Serology must be an integral component of the assessment strategy for intervention trials to control ZIKV. Serologic endpoints may be the most attractive choice for primary endpoints in some settings, particularly in vector control trials, as large numbers of participants could be efficiently and systematically assessed serologically. In other cases, serology may play an adjunct role to refine the sensitivity and specificity of molecular diagnostics. Even if not employed as a primary or secondary endpoint, serologic study of participants is critical for vaccine development. Differences in outcomes among treated subjects permits investigation of differences in immune responses that may enable identification of correlates of protection [203,204,205]. For example, serologic assessment of subgroups of vaccinees has been very informative for DENV vaccine development [206,207]. It is not possible to prescribe a one-size-fits-all approach to serologic testing in all ZIKV intervention trials; however, the features of various assays and testing approaches highlighted here can hopefully inform investigators in constructing efficient and effective testing algorithms that best serve the needs of various trial types and settings. Key questions and the implications for test selection and implementation are provided as a resource in Table 2.

The tools to assess a variety of epidemiologic endpoints in intervention trials to reduce infection or disease caused by Zika virus remain imperfect. However, sufficient experience is accumulating to enable effective implementation of existing serologic tests to answer key questions in the field. It is critical that trial design include close attention to context and trade-offs. Full consideration of how, when, and where trial participants are encountered will guide the testing approach. Trade-offs may be necessary to ensure that important trials are feasible and primary questions and objectives are met even if that means accepting that the most accurate and precise epidemiologic data will not be obtained due to cost, resource allocation, or technological limitations. Additionally, a refined picture of the global epidemiology of Zika infection will also improve the way serologic tests are used in different populations. For example, the seroprevalence of Zika in Southeast Asia is obscure, but it is increasingly appreciated that unrecognized endemic circulation has likely been established for many years in that region. Finally, lessons learned from efforts with Zika can be instructive when faced with re-emerging or novel viruses that share features of basic virology or transmission ecology with Zika.

## Figures and Tables

**Table 1 tropicalmed-04-00068-t001:** Salient features of serologic assays for diagnosing ZIKV infection and considerations for assay selection.

Serologic Basis	Assay	Mechanism and Output	Advantages	Limitations	Utility in Zika Intervention Trials
IgM	MAC-ELISA	Captures human IgM from serum or other fluid (CSF), tests for binding to virus or VLP. Result expressed as P/N ratio.	Substantial experience with assay type Clear guidance on assigning positive and negative results	Flavivirus cross-reactivity reduces specificity IgM may persist beyond acute phase infection Requires local lab optimization	Most useful in detecting acute symptomatic Zika Paired testing of acute and convalescent samples that demonstrates seroconversion or increasing signal is most consistent with acute ZIKV infection A single positive result is typically only supportive of a Zika diagnosis A single positive result from a fetal or neonatal specimen is presumably very useful and specific for ZIKV infection
	In-house	Platform may vary. Can arrange as IgM capture, Ag capture, or direct Ag coating. Output could be P/N or continuous output of background-subtracted OD	May be less expensive than commercial options Can accommodate Ag substitution in the same platform Can give a magnitude of positivity, not simply a binary result	Unlikely to be robustly validated for clinical trial use Similar limitations as MAC-ELISA Methods may not be widely reproduced or adopted by other similar studies	Similar implementation considerations as MAC-ELISA Likely most useful in smaller trials assessing symptomatic Zika infection OR when IgM results are part of secondary outcomes measures rather than the primary outcome
	Kits	Prefab buffers and plates used. Ag may be prM/E or NS1. Readouts typically categorical if not binary based on simple colorimetric reading.	Easy to use Higher throughput vs MAC-ELISA Standardized interpretation Deployable to most lab and field settings	Each kit has to be considered individually Only a few assays with EUA level FDA approval May be costly May not be extensively evaluated for populations where multiple flaviviruses are endemic	Good option for high throughput IgM testing with consistent quality of acute symptomatic subjects (not for testing paired acute-convalescent specimens) Good option for on-site testing in resource-limiting settings Must give attention to how quickly maximum sensitivity is achieved after symptom onset Ultimate case designation should still meet other criteria (as true for other IgM-based tests)
IgG	In-house	Platform may vary. Can arrange as IgG capture, Ag capture, or direct Ag coating. Output could be P/N, typically continuous output of background-subtracted OD	May be less expensive than commercial options Can accommodate Ag substitution in the same platform Can give a magnitude of positivity, not simply a binary result Also can determine total Ab titer to assay Ag	Unlikely to be robustly validated for clinical trial use Flavivirus cross-reactivity reduces specificity	Particularly useful in large scale assessment of incident ZIKV infections (seroconversions) in areas with low ZIKV and DENV seroprevalence IgG seroconversion may be an efficient method for identifying inapparent ZIKV infections Paired testing of acute and convalescent samples that demonstrates seroconversion or increasing signal is consistent with acute ZIKV infection Data on magnitude or Ab titer can be useful in investigating correlates of immunity
	Kits	Prefab buffers and plates used. Ag may be prM/E or NS1. Readouts typically categorical if not binary based on simple colorimetric reading.	Easy to use High throughput similar to In-house Standardized interpretation Deployable to most lab and field settings	Flavivirus cross-reactivity reduces specificity Only a few assays with EUA level FDA approval May be costly May not be extensively evaluated for populations where multiple flaviviruses are endemic	In contrast to IgM kits, IgG kits may be good option for testing paired acute-convalescent specimens to support assessment of symptomatic Zika, though would not be adequate for “lab confirmation” Good option for on-site testing in resource-limiting settings, but these assays are generally not designed or validated for seroprevalence determination in late convalescence, limiting their utility
	Novel IgG	Variable “Novel” here refers to assays that leverage technologic advances	May increase specificity, throughput, automation, reduce sample volume May use multiplexing to simultaneously assess several outcomes	Not widely validated May still be costly or require sophisticated lab capacity/analytic equipment (though there are serious efforts to engineer low costs tests deployable to resource-limited settings)	No novel tests with sufficient development and translational research to favor use in clinical trials currently These assays may constitute key methodologic advances that warrant implementation in near future, perhaps sooner in smaller trials and pilots
IgM or IgG	PRNT FRNT	Measures the functional activity of serum to inhibit live virus infection of target cells. An Ab titer is calculated (i.e., FRNT50)	Most accepted method for serologic diagnosis of flaviviruses Greater specificity compared to IgM and IgG assays Gives information about the flavivirus exposure history, may differentiate primary and secondary flavivirus infections	Requires BLS-2 and culturing infectious virus Labor intensive Several days turn-around time A number of assay components and conditions lead to large lab-to-lab variation Specificity may be reduced in secondary flavivirus infection, particularly in early convalescence	Likely most appropriate to start with neutralization testing to assess convalescent serostatus in areas with high DENV or ZIKV seroprevalence Paired testing of interval samples can detect inapparent ZIKV infection with good specificity for distinguishing from DENV Data on neutralization titer can be very useful in investigating correlates of protection (neutralizing Ab are often hypothesized to be the mediators of lasting protective immunity against ZIKV infection)
	NS1 BOB	Measures the ability of Ab in serum to block the binding of a ZIKV NS1-specific mAb. Readout is percent blockade (0–100%), but initial report includes an ROC curve analysis for binary outcome	High sensitivity and specificity for ZIKV Deployable to most lab settings	Insensitive in 1st week of symptoms Unknown long-term sensitivity (i.e., could there be seroreversion after 1 or more years?)	Could be very useful in assessing recent ZIKV infection in seroprevalence/sero-incidence studies Can be used for testing paired acute-convalescent specimens to serologically diagnose symptomatic Zika May not have sufficient validation for primary outcome measure, but could be efficient and broadly deployable assay for assessing secondary outcomes Particularly useful in monitoring for ZIKV infection in vaccine trials in which subject are immunized with vaccines that lack NS1
	Novel neut	Variable “Novel neut” here refers to neutralization assays that leverage technologic advances	May allow neutralization testing without requiring live virus May improve throughput and automation May be less challenging to standardize than PRNT/FRNT	May require specialized lab capacity or analytic equipment Limited validation to date May have similar reduced specificity in secondary and early convalescent flavivirus infection	Novel neutralization tests could enter trial design strategies more quickly than IgG assays. Validation that the methods give results that are consistently relatable to PRNT/FRNT titers may be sufficient to favor implementation without extensive testing in large populations

ZIKV, Zika virus; ELISA, enzyme-linked immunosorbent assay; IgM, immunoglobulin M; IgG, immunoglobulin G; CSF, cerebrospinal fluid; P/N, a ratio of the optical density of a test sample reacted with ZIKV antigen divided by that of the negative control; prM/E, precursor membrane protein/envelope protein; NS1, nonstructural protein 1; EUA, emergency use authorization; FDA, Food and Drug Administration; DENV, dengue virus; Ab, antibody; Ag, antigen;BSL-2, biosafety level-2; PRNT, plaque reduction neutralization test; FRNT, focus reduction neutralization test; FRNT50, calculated serum dilution at which 50% of maximal neutralization activity occurs; NS1 BOB, NS1 blockade of binding; ROC, receiver operating characteristic.

**Table 2 tropicalmed-04-00068-t002:** Key questions to guide optimal selection and implementation of serologic tools in ZIKV intervention trials. DBS, dried blood spots.

Question	Implication
What is the mechanism of the intervention?	Different vaccine platforms elicit different specificities of Ab responses; these could be to the whole virus, nonstructural proteins, envelope or envelope subdomains only. An intervention on the environment would have no direct effect on ZIKV immunity at the individual level. Vaccines, drugs, immunotherapeutics may have various effects at the level of the host-pathogen interaction and include protective (potentially sterilizing) immunity or attenuate viral infection and/or symptoms. Interventions that target the vector will reduce total infection and a corresponding proportion of disease because symptomatic infection is by definition a subset of total infection.
What is the ultimate goal of the intervention?	The goal may be to reduce disease manifestations, prevent apparent cases, or reduce transmission in the population. The goal may be to prevent infection from ever occurring or to reduce symptoms and risk of complication in a subject after Zika has already been diagnosed.
Who are the key stakeholders and what is their priority?	Stakeholders may include the local community, scientific community, health ministry, trial sponsor, or product developer. Some may prefer to see study results that reflect high confidence and specific diagnosis of each infection. Others may prefer relaxed specificity and greater sensitivity to assess a “bigger picture” of all *Aedes*-borne virus transmission. Disease endpoints as assessed by a physician or a nurse may carry much more validity in the perspective of some stakeholders, whereas others would accept or prefer total infection endpoints that are determined entirely by laboratory testing or even data augmentation or mathematical modeling.
Who is the study population?	A highly mobile population may not work well in a cluster design. A transient population would not be amenable to frequent longitudinal sampling. A pediatric population may have increased retention and sampling compliance if using noninvasive techniques rather than phlebotomy. A study population with high rates of flavivirus vaccination (such as yellow fever or Japanese encephalitis virus) may have increased rates of cross-reactive Ab.
What is the epidemiology at the trial site?	Knowledge of historic flavivirus transmission and likelihood of adequate event rates facilitate planning for optimal serologic testing and design of testing algorithms. If DENV is endemic in the region, assays readily compromised by cross-reactivity, such as those using whole ZIKV antigen, will not be useful. In low seroprevalence settings, ZIKV seroconversion on simple serologic assays can be a robust outcome measure. An area with a high event rate can achieve adequate power with fewer subjects and a shorter trial duration. Additionally, the statistical analysis plan may better account for potential confounding effects of imperfect sensitivity and specificity of serologic assays in this setting.
What resources are available?	Lab capacity for multiplex, high throughput testing and high volume of assays could increase the quality of serologic data generated. However, it may also be reasonable to use less frequent sampling and less expensive assays (such as in-house ELISA and neutralization tests) to preserve funding for other essential trial activities such as clinical evaluations, stakeholder engagement, entomologic monitoring, etc. If specimen transport and cold chain is an issue, DBS is an attractive method to sample large populations.

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
