# Peer review of "Serologic Tools and Strategies to Support Intervention Trials to Combat Zika Virus Infection and Disease"

_tropicalmed, 2019, doi:10.3390/tropicalmed4020068_

Round 1
Reviewer 1 Report
this was most worthy of publication, as an important contribution to the field. Very minor typographical edits could be made. This manuscript should be required reading for persons involved in vaccine development, diagnostics, and public health particularly because the role of IgG was delineated. The field has overemphasized older technologies that have some serious faults when applied to flaviviruses. The author should be commended for his significant contribution in making such a good synthesis of the available literature, the issues regarding flavivirus immunity, and potential role of newer assays and technologies.Author Response
Reviewer 1
this was most worthy of publication, as an important contribution to the field. Very minor typographical edits could be made. This manuscript should be required reading for persons involved in vaccine development, diagnostics, and public health particularly because the role of IgG was delineated. The field has overemphasized older technologies that have some serious faults when applied to flaviviruses. The author should be commended for his significant contribution in making such a good synthesis of the available literature, the issues regarding flavivirus immunity, and potential role of newer assays and technologies.
I appreciate the kind comments of Reviewer 1. I have revised the text to correct minor typos and grammatical errors.
Reviewer 2 Report
Serologic tool and strategies to support intervention trials to combat Zika virus infection and disease
Introduction:
1. Please, go through more carefully the part of ZIKV and probability of enhanced disease. Is this true or not? What is the current opinion about that? Author just lists the DENV and enhanced disease upon infection with a second DENV serotype. ADE part is only discussed in Intervention to control Zika Chapter.
2. Aedes aegypti = Stegomyia aegypti, nowadays. Please, correct that.
3. Introduction, line 48, add period.
Interventions to control Zika virus:
Line 85. Is ADE and ZIKV (pathogenesis etc) in line with DENV? What about pregnant women and ADE (DENV? ZIKV?)? One of the main targets is the pregnant women…
Trial design and outcomes of interest:
Line 103: “…detection of dengue-specific genomic RNA… “ should there be ZIKA-specific genomic RNA?
Line 104: Do author really thinks that detection of ZIKV NS1 antigen/protein is a proper way to diagnose ZIKV infection? If yes, please indicate the studies and data describing and supporting that.
The crucial role of serology in diagnosing ZIKV:
Lines 154-156: In endemic areas where Stegomyia aegypti is one of the common species. The detection of Abs against foreign proteins found in insect saliva, is not giving anything extra information which could help in diagnosis or in vaccine development. You can get Abs easily before ZIKV infection. What the author mean that there is clear advantages in cost and efficiency? To whom? And how?
Natural history of ZIKV infection and host immune response:
Lines 165-169: You can, especially, find prolonged viremia or RNAemia in serum or whole bood samples during pregnancy if fetus is infected with ZIKV (Driggers et al . 2016, NEJM). This has been proved by others as well. This prolonged RNAemia and pregnancy should be mentioned and described here.
Concepts for diagnosing ZIKV infection and disease:
Line 196: Delete virologic (sometimes called molecular) -> just use word molecular there. ...accomplished by molecular or serologic methods.
Available serologic tests for ZIKV and optimal utilization:
Lines 275-280:
This part is confusing. Please, rewrite and modify this. How asymptomatic ZIKV infection could lead to misclassifying an unrelated fever or rash illness if it is asymptomatic? Again, please discuss prolonged ZIKV viremia and/or RNAemia (there is several published studies describing the suitable sample types for screening, not all are good, these references are missing here) , pregnancy and sexually transmitted part in diagnostic concept. Positive RT-PCR result may not be false positive at all. As I said earlier, this part is pretty confusing, what author wants to state here? Molecular methods are reliable methods and maybe the only methods which can discriminate DENV and ZIKV clearly from each other. How about double infection of DENV and ZIKV? These are rare but not that rare after all in endemic areas. How these would be detected in serology or using molecular methods? What are the quidelines for these? Should each patient cohorts be screened by RT-PCR and serologic methods before and after vaccine trials and if symptoms are noticed?
Author Response
Reviewer 2
Introduction:
1. Please, go through more carefully the part of ZIKV and probability of enhanced disease. Is this true or not? What is the current opinion about that? Author just lists the DENV and enhanced disease upon infection with a second DENV serotype. ADE part is only discussed in Intervention to control Zika Chapter.
I removed the clause about enhanced DENV infection, as this is extraneous to the introduction.
I make a more clear statement about the concern for and lack of evidence for ADE playing a role in human Zika infection.
2. Aedes aegypti = Stegomyia aegypti, nowadays. Please, correct that.
Stegomyia has been added in parentheses to the first occurrence to reflect the equality pointed out by the reviewer.
3. Introduction, line 48, add period.
done
Interventions to control Zika virus:
Line 85. Is ADE and ZIKV (pathogenesis etc) in line with DENV? What about pregnant women and ADE (DENV? ZIKV?)? One of the main targets is the pregnant women…
I make a more clear statement about the concern for and lack of evidence for ADE playing a role in human Zika infection. I also state a role of serologic studies in ongoing assessment of safety for Zika vaccines given the experience with dengue vaccines recently.
Trial design and outcomes of interest:
Line 103: “…detection of dengue-specific genomic RNA… “ should there be ZIKA-specific genomic RNA?
Error corrected.
Line 104: Do author really thinks that detection of ZIKV NS1 antigen/protein is a proper way to diagnose ZIKV infection? If yes, please indicate the studies and data describing and supporting that.
Yes. Zika-specific NS1 antigen detection would be a valid way to diagnose symptomatic Zika infection. A discussion on the test parameters and appropriate implementation of this assay is beyond the scope of this review, as this would be considered a molecular method and not a serologic tool. However, serologic work tells us that Zika NS1 is sufficiently divergent from DENV NS1 that a specific assay could be developed, validated and implemented. A reference to one very promising NS1 antigen assay is now included. As with PCR, a positive result with a specific assay should have a high PPV for Zika infection and be useful in a trial setting. A negative result may not rule out infection, thus, my argument for complimentary serologic testing in the context of diagnosing symptomatic cases.
The crucial role of serology in diagnosing ZIKV:
Lines 154-156: In endemic areas where Stegomyia aegypti is one of the common species. The detection of Abs against foreign proteins found in insect saliva, is not giving anything extra information which could help in diagnosis or in vaccine development. You can get Abs easily before ZIKV infection. What the author mean that there is clear advantages in cost and efficiency? To whom? And how?
A few changes have been made to clarify. First, this section has been broken into two paragraphs, the first deals entirely and explicitly with diagnosing Zika infection, which is the primary use of serology being discussed in this article. The second paragraph explores supplemental advantages of serology, one of which includes indirect methods of assessing vector exposure burden (admittedly, the techniques are in early stages of investigation and development). The cost and efficiency gains come in leveraging specimens generated in assessment of primary objectives to accomplish secondary or exploratory objectives. Specifically for serology to estimate mosquito bite burden, this alternative approach could save substantial cost associated with entomologic monitoring and it could give an idea of exposure intensity, which can be heterogeneous over relatively small geographic scales. It is true that most subjects in an area will likely have been exposed to mosquito bite, but resulting serologic responses may be transient and/or vary in magnitude depending on when and how many bites received.
Natural history of ZIKV infection and host immune response:
Lines 165-169: You can, especially, find prolonged viremia or RNAemia in serum or whole bood samples during pregnancy if fetus is infected with ZIKV (Driggers et al . 2016, NEJM). This has been proved by others as well. This prolonged RNAemia and pregnancy should be mentioned and described here.
Reference added
Concepts for diagnosing ZIKV infection and disease:
Line 196: Delete virologic (sometimes called molecular) -> just use word molecular there. ...accomplished by molecular or serologic methods.
Deleted
Available serologic tests for ZIKV and optimal utilization:
Lines 275-280:
This part is confusing. Please, rewrite and modify this. How asymptomatic ZIKV infection could lead to misclassifying an unrelated fever or rash illness if it is asymptomatic? Again, please discuss prolonged ZIKV viremia and/or RNAemia (there is several published studies describing the suitable sample types for screening, not all are good, these references are missing here) , pregnancy and sexually transmitted part in diagnostic concept. Positive RT-PCR result may not be false positive at all. As I said earlier, this part is pretty confusing, what author wants to state here? Molecular methods are reliable methods and maybe the only methods which can discriminate DENV and ZIKV clearly from each other. How about double infection of DENV and ZIKV? These are rare but not that rare after all in endemic areas. How these would be detected in serology or using molecular methods? What are the quidelines for these? Should each patient cohorts be screened by RT-PCR and serologic methods before and after vaccine trials and if symptoms are noticed?
This section has been extensively revised following this feedback. The main attempt to improve and clarify the section is to focus first on the available tests with a bit more objectivity, and then discuss advantages and disadvantages of test implementation in the frame of a few trial-specific contexts.
Reviewer 3 Report
This paper is on an important and interesting topic and I would very much encourage the author to develop it for publication. However, I do feel at the moment that it is much too long and contains a great deal of unnecessary information - indeed it takes until page 5 to reach the main topic of serologic tools. Despite being too long, the paper does not provide any conclusion and will not help any researcher to select appropriate serologic tests for her intervention trial.
The promises in the sentence in the abstract (lines 18 and 19) are not fulfilled. This is disappointing.
There are a lot of typos and grammatical errors that need correction and thorough proof reading. The author needs to appreciate the difference between principle and principal.
More specific comments:
The sentence on lines 88-89 needs an expanded explanation.
The para lines 101- 114 is confusing. is this about ZIKV or DENV?
The test-negative trial design needs to be explained more clearly (Lines 117-119).
Why is the case fatality rate for Ebola relevant to this topic (lines 124-5)?
The NS1 BOB assay needs an expanded description and referencing (lines 255-257).
Is the neutralisation testing mentioned in lines 262-263 PRNT? If so, say so, as the general reader will recognise this abbreviation.
What is meant by 'vector-based trials'? Lines 319-320.
Author Response
Reviewer 3
This paper is on an important and interesting topic and I would very much encourage the author to develop it for publication. However, I do feel at the moment that it is much too long and contains a great deal of unnecessary information - indeed it takes until page 5 to reach the main topic of serologic tools. Despite being too long, the paper does not provide any conclusion and will not help any researcher to select appropriate serologic tests for her intervention trial.
The promises in the sentence in the abstract (lines 18 and 19) are not fulfilled. This is disappointing.
I appreciate Reviewer 3’s statement and encouragement. These points are well taken. I have extensively revised the manuscript to incorporate this feedback and improve the manuscript. Because this involves substantial editing and text rearrangement, I have accepted many changes so the entire manuscript is not red.
The key changes include reorganization of major points.
- the section on natural history and immunity has been shortened and combined with the section that discusses basic concepts of serologic detection and diagnosis of Zika. This shortens the manuscript and gets to the main topic of serology by page 2. To further increase the efficiency (shorten) of the discussion, information and discussion of molecular diagnostics is substantially reduced.
Finally, a “synthesis” paragraph is added after discussion of both serologic tools and trial design, to give specific examples and suggestions for how to implement serologic tools effectively – with the goal of fulfilling the promises of the abstract.
There are a lot of typos and grammatical errors that need correction and thorough proof reading. The author needs to appreciate the difference between principle and principal.
I have revised the text to correct minor typos and grammatical errors. The two misuses of the homonyms principle vs principle were corrected.
More specific comments:
The sentence on lines 88-89 needs an expanded explanation.
Sentence edited to refer more specifically to the background knowledge available. Discussion of “new technologies” is left until the later dedicated section.
The para lines 101- 114 is confusing. is this about ZIKV or DENV?
A clause was added in line 101 to clarify that the statement is about ZIKV. Later in the same paragraph, DENV is one of the “other arboviruses” to which some of the concepts discussed in the article apply. DENV is also frequently discussed as a point of reference, comparator, or example throughout the article as the literature on DENV is much more robust than for ZIKV.
The test-negative trial design needs to be explained more clearly (Lines 117-119).
Paragraph revised for clarity.
Why is the case fatality rate for Ebola relevant to this topic (lines 124-5)?
The allusion to Ebola and the trial ethics questions that arise was deleted, as this is extraneous to the main point of this article.
The NS1 BOB assay needs an expanded description and referencing (lines 255-257).
An additional explanatory sentence and reference has been added.
Is the neutralisation testing mentioned in lines 262-263 PRNT? If so, say so, as the general reader will recognise this abbreviation.
I prefer to refer to “neutralization testing” independent of the method to discuss the concept. I have edited the appropriate section to specifically name PRNT and FRNT as the most common form of neutralization assays.
What is meant by 'vector-based trials'? Lines 319-320.
I changed the term to ‘vector control,’ which should be better understood.
Round 2
Reviewer 2 Report
No more comments.
Author Response
The reviewer indicates no further comments.
I thank the reviewer for their time in assessing the 2 previous versions of the manuscript.
Reviewer 3 Report
This paper is much improved now, so many thanks to the author. It is much more focused on the topic and consequently more readable. The author has also clarified may of the issues in the previous version.
I have one main suggestion. I think this paper would be made much more useful to readers if the authro could add a table listing the main available tests, what they measure, when they can be useful and what are their main drawbacks. At the moment this is covered in the text, but is not very accessible for readers. A summary of this in a table would help and this would greatly complement the current table which lists the considerations for selecting a test.
I other comments are minor:
Page 3 para 2 line 6: DPO - is this days post onset or decline post onset? On the same line the text reads '2 weeks DPO' while in para 4 we have '10-15 DPO'. Hence my uncertainty. One of these must be wrong.
Page 3 para 3 line 8: 'supportive but definitive...' does not make sense. Is a 'not' missing?
Page 3 para 4, line 9: ZIVK is a typo.Should be ZIKV.
Page 3 para 5 line 3 - what is PrM? I cann;t see this abbreviation explained anywhere.
Page 4 - IgG test are generally more problematic than IgM for cross reactivity with zika. However, this does not currently come across from the text. In fact the casual reader might well conclude IgM tests are more problematic. I feel this needs a bit of a rewrite.
Page 5 para - is this describing a specific trial rather than the test negative design? This needs clarifying as the text seems to be about DENV rather than ZIKV. If a specific trial, then I suggest the author mentions this as an example of the method and comment that this method could also be used for ZIKV.
Author Response
Reviewer 3
This paper is much improved now, so many thanks to the author. It is much more focused on the topic and consequently more readable. The author has also clarified may of the issues in the previous version.
I have one main suggestion. I think this paper would be made much more useful to readers if the authro could add a table listing the main available tests, what they measure, when they can be useful and what are their main drawbacks. At the moment this is covered in the text, but is not very accessible for readers. A summary of this in a table would help and this would greatly complement the current table which lists the considerations for selecting a test.
I appreciate this point. I have included a table and tried to address all of the recommendations. I do want to avoid giving a specific endorsement of any one test or assay, particularly commercial products, so nothing that direct and specific will appear in the table. My main goal is to point out many of the key considerations to aid researchers in developing the testing approach that best fits their objectives and capacity.
I other comments are minor:
Page 3 para 2 line 6: DPO - is this days post onset or decline post onset? On the same line the text reads '2 weeks DPO' while in para 4 we have '10-15 DPO'. Hence my uncertainty. One of these must be wrong.
Days post onset of symptom (DPO) is the correct version. Corrected the “2 weeks.” I see DPO used more frequently for clinical studies with dengue and now Zika (rather than DPI for days post infection that is more often used in experimental infection models) since that is usually the earliest event that can be clearly known. The precise day of human ZIKV infection could fall anywhere in the incubation period.
Page 3 para 3 line 8: 'supportive but definitive...' does not make sense. Is a 'not' missing?
Corrected
Page 3 para 4, line 9: ZIVK is a typo.Should be ZIKV.
Corrected
Page 3 para 5 line 3 - what is PrM? I cann;t see this abbreviation explained anywhere.
Now defined on 1st occurrence
Page 4 - IgG test are generally more problematic than IgM for cross reactivity with zika. However, this does not currently come across from the text. In fact the casual reader might well conclude IgM tests are more problematic. I feel this needs a bit of a rewrite.
I have added an introductory sentence for Page4 para 2 to this effect. I also think the table addresses this point to some extent.
Page 5 para - is this describing a specific trial rather than the test negative design? This needs clarifying as the text seems to be about DENV rather than ZIKV. If a specific trial, then I suggest the author mentions this as an example of the method and comment that this method could also be used for ZIKV.
I have specifically named the AWED trial as an example of this trial design.
Round 3
Reviewer 3 Report
Thanks to the author for being so open and constructive in engaging with referees' comments. I have no further comments.